# Potential of 3D Hierarchical Porous TiO_2_-Graphene Aerogel (TiO_2_-GA) as Electrocatalyst Support for Direct Methanol Fuel Cells

**DOI:** 10.3390/nano13121819

**Published:** 2023-06-07

**Authors:** Siti Hasanah Osman, Siti Kartom Kamarudin, Sahriah Basri, Nabila A. Karim

**Affiliations:** 1Fuel Cell Institute, Universiti Kebangsaan Malaysia, UKM, Bangi 43600, Selangor, Malaysia; 2Department of Chemical and Process Engineering, Faculty of Engineering and Built Environment, Universiti Kebangsaan Malaysia, UKM, Bangi 43600, Selangor, Malaysia

**Keywords:** titanium dioxide, graphene aerogel, catalyst support, direct methanol fuel cell

## Abstract

Fuel cells have already demonstrated their potential for green energy generation. However, the low reaction performance becomes an obstacle in terms of large-scale commercial manufacturing. Accordingly, this work focuses on a new unique fabrication of three-dimensional pore hierarchy TiO_2_-graphene aerogel (TiO_2_-GA) supporting PtRu catalyst for anodic catalyst direct methanol fuel cell, which is facile, ecologically benign, and economical. In this work, a hydrothermal technique was used, followed by a freeze-drying technique and a microwave-assisted ethylene reduction technique. The structural properties of the studied materials were confirmed by UV/visible spectroscopy, XRD, Raman spectrum, FESEM TEM, and XPS. Based on existing structural advantages, the performance of PtRu/TiO_2_-GA has been investigated on DMFC anode catalysts. Furthermore, electrocatalytic stability performance with the same loading (~20%) was compared to commercial PtRu/C. Experimental outcomes show that the TiO_2_-GA support offered a significantly high surface area value of 68.44 m^2^g^−1^, mass activity/specific activity (608.17 mAmg^−1^/0.45 mA/cm^2^
_PtRu_) that is higher than commercial PtRu/C (79.11 mAmg^−1^/0.19 mA/cm^2^
_PtRu_). In passive DMFC mode, PtRu/TiO_2_-GA showed a maximum power density of 3.1 mW cm^−2^, which is 2.6 times higher than that of the PtRu/C commercial electrocatalyst. This suggests that PtRu/TiO_2_-GA has a promising possibility for methanol oxidation and may be used as an anodic element in DMFC.

## 1. Introduction

Fuel cells have been developed as energy conversion technologies in recent decades, in response to rising environmental pollution and developing global energy problems [1]. As a result, research regarding the advantages of the direct methanol fuel cell (DMFC) over alternative fuels has received a lot of attention. Liquid methanol can be created with high energy conversion efficiency coefficients, a narrow temperature working range, and low pollution levels, and liquid methanol should be stored and transported in an ecologically safe manner [2]. The most efficient catalyst is platinum (Pt), which is applied for hydrogen reactions (HOR) and oxygen reduction reactions (ORR). However, Pt is expensive, so researchers are looking for other alternatives to find comparable new catalysts for electrocatalyst applications [3,4]. Therefore, in the catalyst, ruthenium is present and can counteract CO poisoning [5]. Consequently, with the use of PtRu alloys, CO poisoning on the catalyst can be avoided, and this is also one of the solutions in the industry as a whole [6,7]. In DMFC applications, it has been recommended to utilize Pt with a standard ratio of 1:1, which can minimize Pt loading while lowering the catalyst cost.

Rhodium (Rh) nanocrystals have emerged as a popular subject of discussion due to their excellent methanol oxidation activity and high resistance to byproducts (mainly CO) in alkaline medium, making them a promising alternative to various Pt-based materials [8,9]. However, like other noble metals, the catalytic performance of Rh nanocrystals relies heavily on their size and morphology. Therefore, it is highly desirable to have a systematic approach for designing and synthesizing small and well-dispersed Rh catalysts. Nevertheless, Ru is an essential bifunctional catalyst in HOR because it can remove carbon monoxide (CO) from these active sites, and the catalyst is no longer CO poisoned [10,11]. According to Kua and Goddard, the dual-function mechanism of the Ru-Pt catalyst plays a role in the oxidation of methanol, where [12] platinum acts as the active catalyst in the dehydrogenation of methanol while ruthenium plays a major role in the dehydrogenation of water. Despite the advantages of PtRu, the problem of low methanol oxidation remains unresolved. PtRu demonstrated higher catalytic activity in the DMFC, corresponding to research by Bock et al., and it was evident that the catalytic performance was significantly dependent on the distribution of Pt and Ru sites at the atomic level [13]. Therefore, more specific changes need to be made to the catalyst to assist the fuel cell industry.

Concerning catalytic components, one of the interesting approaches among researchers has been the overview of metal oxides and nanomaterials. Despite being recognized as an effective semiconductor catalyst, there is a relative scarcity of literature regarding the photocatalytic activity of pure MnO_2_ [14], but TiO_2_ has received tremendous attention in the last few decades because of its nontoxic properties and excellent optical, photocatalytic, and electrical properties [15,16,17]. Instead, titanium dioxide (TiO_2_) is one part of metal oxides, often known as titania, used to improve the electrocatalysis of DMFCs. Ercelik and his team [18] studied the Pt-Ru/C-TiO_2_ anode electrocatalyst characterization and performance evaluation for DMFC purposes. Regarding the increased reactivity at the surface, anatase charge carriers are more efficient in bulk materials and have a beneficial impact on catalytic activity [19]. TiO_2_ is an inorganic substance that is stable, non-flammable, and corrosion resistant. The interaction of TiO_2_ metal oxide with other materials, in general, can improve the electronic behavior of materials. Consequently, in this instance, the oxidation activity can be increased while the oxidation of CO can be decreased [20]. The kinetic reactions and reaction mechanisms that take place can be influenced by the use of TiO_2_ as a supporting substance in metal catalysts [21].

Surprisingly, novel carbon supports such as graphene aerogel (GA) have recently developed in the evolution of 3D graphene, capturing the interest of researchers due to their unique qualities such as having higher electrical conductivity, transport efficiency than graphene, and generating an extreme surface area, followed by facilitating the deposition of nanostructured materials. Zhao et al. [22] introduced a high-stability electrocatalyst for methanol electrooxidation using a hybrid of carbon-supported Pt nanoparticles and three-dimensional graphene aerogel. Graphene aerogels feature a light, large surface area, outstanding mechanical elasticity, and excellent conductivity due to linked 3D mesopores and micropore structures [23,24]. Accordingly, a flexible combination of the rational structure of TiO_2_ and graphene aerogels, porous nanostructures, and catalytic structures is extremely beneficial in boosting the overall methanol oxidation performance potential [25]. 

Some research recently revealed progress in reaction kinetics and performance of the DMFC by different support in half cells, especially demonstrated by Basri and colleagues [26] proposed the novel multi-component anode catalyst, PtRuFeNi/MWCNT (31 mA mg^−1^), while Abdullah and team [27] introduced PtRu/TiO_2_-CNF (345.64 mA mg^−1^) and Ramli et al. [11] studied PtRu/CNF (427 mA mg^−1^) as anodic catalyst support. The finding of TiO_2_-GA in this study gives high electrocatalyst performance compared to nanocomposites that have been studied previously based on the characteristics and properties of nanocomposites themselves. Based on the electrocatalyst performance of this new composite, it is better than the PtRu catalyst.

Apparently, this is the first work that described the study that combines the hydrothermal and freeze-drying methods to synthesize and characterize PtRu catalyst with novel TiO_2_-GA support. According to the TEM findings, PtRu/TiO_2_-GA was uniformly dispersed at all surface angles. Based on electrochemical tests, it was discovered that the PtRu/TiO_2_-GA catalyst displayed greater electroactivity than the other catalysts. Furthermore, PtRu/TiO_2_-GA catalysts provide a higher power density in direct methanol fuel cell (DMFC) passive single cells compared to commercial PtRu/C. Hence, this novel electrocatalyst can be utilized and potentially discovered in a low-temperature fuel cell.

## 2. Experimental Section

### 2.1. Materials

Titanium isopropoxide (TiPP, 97%) was obtained from Sigma-Aldrich Co., Burlington, MA, USA, and graphene oxide was obtained from GO Advanced Solution Sdn. Bhd., Kuala Lumpur, Malaysia. The Pt precursor, H_2_PtCl_6_, with 40% content, was kindly supplied by Merck, Darmstadt, Germany, while the Ru precursor, RuCl_3_ (45–55% content), supplied by Sigma-Aldrich Co., was used in the synthesis catalyst. Ethylene glycol (EG), Nafion solution, isopropyl alcohol, ethanol, and methanol were obtained from Sigma-Aldrich.

### 2.2. Preparation of TiO_2_-GA Composite

TiO_2_-GA was synthesized using hydrothermal and followed by a freeze-drying technique. Firstly, 20 mg of TiO_2_ were homogenously moderated by ultrasonication for 2 h and distributed into 40 mL of GO suspension (2 mg/mL). After that, the solution was transferred to a 50 mL Teflon-lined autoclave and kept at 200 °C for 12 h. The TiO_2_-graphene hydrogel was completely washed with deionized water before being freeze-dried for 24 h to yield a TiO_2_-graphene aerogel (TiO_2_-GA) composite. The same procedure was used to make pure graphene aerogel (GA) for the control experiment.

### 2.3. Preparation for the Synthesis of PtRu/TiO_2_-GA Catalyst

The synthesis was determined according to the standard atomic ratio of 1:1 of PtRu catalyst with 20 wt% loaded on TiO_2_-GA support. The synthesis began with weighing chloroplatinic acid (Pt source) and ruthenium chloride (Ru source) precursors, followed by mixing with ethylene glycol (EG) solutions sonicated for 15 min. After homogeneously mixed well, TiO_2_-GA was added to the precursor solution by stirring for about 30 min with adjusting the pH solution to 10 using 1 M NaOH solution. To guarantee the reduction process was complete for this synthesis, the microwave-assisted ethylene reduction technique was introduced for 1 min and then turned off for 1 min twice. Finally, the composite was filtered and washed a few times with ethanol and DI water before being dried in the oven at 120 °C for 3 h.

### 2.4. Physical Characterization

For the determination of the physical characteristics, optical properties such as UV/visible spectroscopy (Perkin Elmer, Lambda 35, UK, JEM-1010 JEOL, Tokyo, Japan) in the 200–800 nm range were used to characterize the samples. X-ray diffraction (XRD) is a potent physical characterization that may be used to demonstrate the crystallinity of manufactured materials. Meanwhile, Raman spectrum analysis was practical to designate the degree of graphitization. Field emission scanning electron microscopy (FESEM) analysis was applied to study the surface morphology and shapes of the samples, while the instrument JEM-1010 JEOL with 100 kV was used for transmission electron microscopy (TEM) analysis.

### 2.5. Electrochemical Measurements

Electrochemical surface area (ECSA) was determined using cyclic voltammetry (CV) in an acid medium of 0.5 M H_2_SO_4_ for this study. Currently, a three-electrode system was employed at room temperature with Ag/AgCl electrodes as the reference electrodes, a counter electrode with platinum contains, and a working electrode with a glassy carbon electrode (3 mm Ø). The addition of 150 μL IPA, 150 μL DI water, and 50 μL Nafion solution (5 wt%) maintained by 2.5 mg electrocatalyst was used to prepare the electrocatalyst ink for the working electrode. Specimens were conditioned at 0.0503 mgcm^−2^ with a catalyst loading of 20% wt PtRu as a test on the glassy electrode. For CV measurement, the potential range of −0.2 to +1.0 V vs. Ag/AgCl and a scan rate of 50 mVs^−1^ were measured using the Autolab electrochemical workstation.

### 2.6. Fabrication of MEA

The Nafion 117 membrane was used in this experiment because it is a commonly used commercial membrane in DMFC technology. To maximize the function of MEA, the membrane pretreatment method is very important to remove all organic impurities bound to the membrane layer [28]. A piece of Nafion 117 membrane measuring 3 cm × 3 cm was placed in a beaker containing 200 mL of DI water and heated to 80 °C for 1 h. At the same temperature and time, the deionized water was replaced with a solution of hydrogen peroxide (H_2_O_2_). After that, the H_2_O_2_ solution is drained from the beaker. The membranes were then washed and soaked in deionized water for 2 h at room temperature. Then, 1 M sulfuric acid (H_2_SO_4_) was poured into a beaker with 117 membrane sheets and heated for 1 h at 80 °C. The membrane was excluded once more and washed with deionized water until it was neutral. Until deionized water is available, membranes should always be weighed in it. Due to their use in acidic media, these membranes are first treated with sulfuric acid. Finally, the MEA was created by sandwiching the anode and cathode electrodes within a commercial MEA (Nafion-N117) using a hot press for 3 min at 135 °C and 50 kPa of pressure. The modified procedures from Zainoodin and colleagues [29] served as the basis for all the parameters in this procedure.

### 2.7. DMFC Performance Test

The outcome of the PtRu/TiO_2_-GA and PtRu/C electrocatalysts in the DMFC was assessed using a galvanostat/potentiostat and polarization graph. A DMFC was used to house the MEA, which has an active area of 4 cm^2^. 10.0 mL of 2.0 M methanol fuel was added to the anode side of the DMFC for testing with the potentiostat/galvanostat. The PtRu/TiO_2_-GA and PtRu/C single-cell tests were carried out at room temperature under passive conditions.

## 3. Results and Discussion

### 3.1. Electrocatalyst Physical Characterizations

TiO_2_-GA composite manufacturing technique is depicted in Figure 1. In composite fabrication, several methods using a one-pot hydrothermal stage and freeze-drying step have been presented. To prevent GO aggregation during the reaction and get products with a large surface and open framework, a one-pot hydrothermal method was used. The developed TiO_2_-graphene hydrogel was produced in cylindrical form after a 200 °C hydrothermal reaction for 12 h, and this condition is appropriate for use in this study. The Teflon mold was used to create this cylindrical shape, and the larger the resulting shape, the greater the volume obtained. Furthermore, by freeze-drying the hydrogel, the re-agglomeration of graphene is prevented, it can be well stored for further physical analysis, and it shows light features with which it can stand on flower pieces.

The optical absorption ability of the electrocatalyst was estimated using UV/visible spectroscopy between wavelengths of 200–800 nm, with PtRu/C as a reference, as illustrated in Figure 2. PtRu/C exhibits a medium yellow absorption peak with UV/vis absorption band characteristics at 250 nm, PtRu/GA around 222 nm, PtRu/TiO_2_ around 332 nm, and PtRu/TiO_2_-GA around 277 nm. The normalized spectrum, whose peak spectrum ≈ 230 nm, can be used to analyze the concentrations of total organic carbon (TOC) [30,31]. In general, the lower the peak position (as a wavelength) the lower the aromatic level of a planar structure such as graphene. Accordingly, the PtRu/GA samples displayed the shortest wavelengths compared to the others. The main peak of approximately 300 nm is due to the transition of ∏→∏*carbonyl groups, which can reflect the concentration of the C=O group (carbonyl or carboxyl) as well as the degree of oxidation [32,33]. The PtRu/TiO_2_-GA samples have a wavelength close to 300 nm and the results of this study are approximate to the report already reported by Huang and the group [34], where the samples having the graphene element have a large side size, hence the hydroxyl reduction at the edge of the aromatic domain and carboxyl groups occurs [32]. Moreover, at 200 °C, the high degree of oxidation is influenced by the high oxidation in the basal plane of graphene, and the number of aromatic rings also increases at 230 nm absorption. PtRu/TiO_2_-GA, on the other hand, demonstrated a greater intensity shift than PtRu/GA. Based on experimental results, that conclusion supports that heating allows further reduction of the oxygenated groups that link GA to the TiO_2_ surface. Furthermore, the increase in GA hydrophobicity at higher reduction temperatures prevents the interaction between GA and the hydrophilic TiO_2_ surface, causing phase separation. This suggests that when TiO_2_ is placed on the GA surface, the catalyst efficiency improves. Due to the chemical reduction that happens between PtRu and TiO_2_-GA ions during electrocatalyst production, it is believed that the electrocatalyst composite can boost the electrocatalyst activity.

XRD analysis was used to study the crystal structure of TiO_2_-GA composites and electrical catalysts. A comparison of XRD patterns based on TiO_2_, GA, TiO_2_-GA, and electrocatalysts is illustrated in Figure 3. X-ray diffractometers were used in this analysis in the range of 5°–90° with 2θ. Enhanced in Figure 3a, GA shows the appearance of a new width at 2θ = 24.4° corresponding to a distance between layers of 0.36 nm. The reduction in the distance between the layers is due to the removal of oxygen-containing functional groups. Peaks located at 25.3°, 37.8°, 48.0°, 53.9°, 55.1°, 62.7°, 68.8°, 70.3°, and 75.0° are indexed to (101), (004), (200), (105), (211), (204), (116), (220) and (215) TiO_2_ anatase crystal planes, respectively [35,36]. The diffraction pattern of TiO_2_-GA is similar to that of TiO_2_ nanoparticles in the XRD results. TiO_2_ nanoparticles effectively adhere to GA during the TiO_2_-GA self-assembly process generated by hydrothermal procedures, in which temperature parameters play a role in the successful transition of TiO_2_ structure from anatase to rutile. This temperature shift occurs at elevated temperatures above 700 °C [36]. However, the temperature used in this investigation was only 200 °C.

Diffraction peak analysis for all relevant electrocatalysts and XRD pattern comparison samples to show the PtRu/TiO_2_-GA electrocatalyst are shown in Figure 3b. The peak for TiO_2_ is almost the same as for the TiO_2_-GA sample (JCPDS No. 21-1272). Pt and Ru stand out for each material; there are four more peaks, namely Pt at 39.7° (111), 46.2° (200), 67.5° (220), and 81.3° (311). The diffraction peaks for Ru are 40.7° (111), 47° (200), 69° (220), and 83.7° (311). The structures of the two metals are so solid that the high Bragg angle is found in electrocatalyst samples in the range of 25°–60°. This suggests that catalysts have bimetallic or alloy interactions [37]. All electrocatalyst samples had weak intensities and wide showed high dispersion in the resulting samples. The Debye–Scherrer equation [26] was used to calculate the crystal size of 0.98α divided by βcosθ, where α is the X-ray wavelength, β is the peak width at half the height, and θ is the angle at the peak. Crystal measurement values for evaluating XRD findings could be obtained using Eva software and calculated using the Debye-Scherrer equation. Crystal measurements for all samples are recorded in Table 1. Particle size variables are critical in catalyst activity to achieve good catalyst performance. The range for all crystal readings was 3.8 to 9.8 nm. These results indicate that electrocatalysts are recognized as nanosized.

This study becomes more intriguing when it is expanded by doing a Raman spectrum investigation to identify the vibrations of the species carbon in the specimen. Figure 4 describes the Raman spectra for all samples by displaying the D and G bands at 1350 and 1590 cm^−1^ by two distinct peaks, respectively, where the D band corresponds to crystal boundary vibration and the G band reflects the vibrations of the graphitic crystal of the perfect sp2. In the analysis, the G band corresponds to the vibration mode of the hybridized carbon atom sp2 on the graphite layer in general [38]. Furthermore, the D band exhibits poor crystallinity for TiO_2_-GA support at broad peaks. Moreover, the value (ID/IG) may be used to assess the value of the defect, with the bigger value (ID/IG), indicating that there are faults in the graphite [39].

The Raman spectrum study provided the following values (I_D_/I_G_) in descending order: TiO_2_-GA (1.02), PtRu/TiO_2_-GA (0.99), PtRu/C (0.98), and PtRu/GA (0.95). Accordingly, this varies based on the relative intensity of the D and G bands on the kind of graphite material that can be used to determine the degree of graphitization. The ratio calculated in this study by (I_D_/I_G_) obtained shows that GO has been successfully reduced in aerogel after the catalyst is dispersed at TiO_2_-GA [40,41]. In conclusion, the carbon layer in these electrocatalysis samples was not significantly different in the decomposed region and structure of TiO_2_-GA. The weak Raman shift bands at 152 cm^−1^ and 154 cm^−1^ were discovered in TiO_2_GA and PtRu/TiO_2_-GA, respectively, which are connected to the special properties of TiO_2_ anatase [42].

XPS analysis was performed on a synthesized composite electrocatalyst, PtRu/TiO_2_-GA, to further understand the chemical and bonding environment of this electrocatalyst. Additionally, XPS analysis also provides quantitative information about the elements by using peak widths on the spectrum [43]. The XPS spectra of the PtRu/TiO_2_-GA electrocatalyst are shown in Figure 5a. As can be seen, the study spectra have proved the presence of the elements Pt, Ru, Ti, N, O, and C in the electrocatalyst samples. Figure 5b–g shows high-resolution XPS spectra for Pt 4f, Ru 3d, Ti 2p, N 1s, O 1s, and C 1s found in the samples.

The analysis results for the high-resolution spectra of C 1s, as shown in Figure 5b, highlight four peaks located at 284.5 eV, 285.6 eV, 286.4 eV, and 286.9 eV. The peak located at 284.5 eV explains the existence of carbon fusion in various organic groups on the sample surface [44]. Moreover, the peaks seen at 285.6 eV, 286.4 eV, and 286.9 eV were C–O bonds and C=C [45].

Figure 5c shows the XPS spectra for the prominent element N 1s, the presence of element N located on the carbon support graphite structure. The high-resolution XPS spectrum also revealed the appearance of binding energy around 400 eV due to the presence of a pyrrolic nitrogen atom component on the carbon support structure. The high catalytic activity of the metal PtRu/TiO_2_-GA can also be attributed to the presence of element N in this electrocatalyst. As previously reported by past research, element N can reduce or overcome the problem of agglomeration between particles, which in turn can be attributed to the increased electrocatalyst activity of MOR [46,47].

As shown in Figure 5d, the appearance of nearby peaks of 78.7 eV and 82.3 eV involved Pt 4f_7/2_ and Pt 4f_5/2_, respectively. These binding energy values represent different levels of Pt oxidation, namely 78.7 eV is Pt metal and 82.3 eV is Pt metal oxide (PtO) species. As can be seen at the top of the binding energy graph, Pt metal (around 55–60%) is the most influential species for Pt 4f. These XPS results suggest that the majority of Pt surface sites are present in an easily reducible form, as required for methanol oxidation reactions [48]. Additionally, the analysis of the XPS results showed that the majority of Pt was in the form of a single metal, namely Pt (0), which confirmed that the reduction method using ethylene glycol solvent successfully reduced Pt from its precursor.

The binding energy for the element Ru is shown in Figure 5e, where Ru 3d_5/2_ and Ru 3d_3/2_ are identified to be approximately 280.3 eV and 284.2 eV, respectively. The difference in the value of the binding energy is 3.9 eV, which is within the normal Ru difference. The XPS spectra for Ru 3d showed the presence of a low Ru component (intermediate between Ru metal and Ru (+IV) -oxide) at the peak of 280.3 eV and the RuO_2_ component at the peak of 284.2 eV. The presence of peak O 1s located at 531.3 eV further strengthens the evidence of the presence of the RuO_2_ component [49].

Figure 5f shows the binding energies of Ti 2p_3/2_ and Ti 2p_1/2_ identified to be approximately 459.5 eV and 465.1 eV, respectively. The separation between Ti 2p_3/2_ and Ti 2p_1/2_ is 5.6 eV, which is consistent with the standard value of normal TiO_2_. The high-resolution spectra of O 1s in Figure 5g yield two peaks at 531.3 eV and 532.7 eV, which illustrate the existence of metal oxide bond species, Ti–O–Ti (O lattice) and Ti–OH, respectively.

The existence of bonds such as C–O/C=O and C–OH present demonstrate the effectiveness of the reduction method well performed, and this is equivalent to the gain on the results of FTIR analysis for TiO_2_-GA catalyst support. Based on the matching analysis of the O 1s and C 1s curves, it was observed that the carbonated polymeric layer bonded to the TiO_2_ surface through Ti–O–C and Ti–OCO bonds.

The percentages of atomic element concentrations for all substances present in the samples are tabulated in Table 2. The Pt and Ru presence ratios obtained from the XPS analysis were 0.65:1, equal to the value of the 1:1 ratio. Therefore, the ratio of this metal catalyst corresponds to the value of the ratio required for this experiment. The support ratio of TiO_2_ and GA catalysts is 1:0.88. Overall, the ratios of the four elements, Pt:Ru:TiO_2_:GA matched the calculated values and expectations of the study.

The morphology of the surface of bimetallic electrocatalyst-based Pt on TiO_2_-GA was investigated further using FESEM analysis with Supra 55 VP. In addition, all prepared electrocatalysts identify the distribution of the elements in each sample in Figure 6. In the ultrathin layer of the aerogel matrix, the composite TiO_2_-GA has a superior porosity structure, as shown in Figure 6a. TiO_2_ content in TiO_2_GA is 12.4 wt%, while in PtRu/TiO_2_-GA is 8.1 wt%, respectively. This corresponds to the Raman shift band, which reveals TiO_2_ in anatase. Additionally, the greater TiO_2_ ratio resulted in the formation of agglomerate TiO_2_ particle bulks of graphene, causing the graphene network to collapse. The evaporation of free water during freeze-drying or moisture obtained from the air causes the initial drop in this TiO_2_ weight. However, there is no clear TiO_2_ particle or aggregation observed in the image of FESEM. The graphene clumps, and the surface of the cell wall appears wrinkled and as though it has become a clump of aerogel, with some holes and crispation in the edge requiring sample treatment. Furthermore, following hydrothermal treatment, surface-anchored TiO_2_ nanoparticle clusters form on the developing TiO_2_ nanoparticles [50]. PtRu/TiO_2_-GA was successfully synthesized through the microwave-assisted ethylene reduction technique in Figure 6b. The PtRu nanoparticles are of diverse shapes with smaller fixed sizes and are widely disseminated. The interface demonstrated that the PtRu nanoparticles have been well submerged in TiO_2_-GA, implying that TiO_2_-GA and PtRu nanoparticle alloys have a strong interaction. This validated PtRu/TiO_2_-GA, in which PtRu nanoparticles are homogeneously distributed and effectively formed onto TiO_2_-GA.

Next, a TEM analysis was carried out to confirm the presence of TiO_2_-GA structural formation, as shown in Figure 6c. TEM images further confirmed that TiO_2_ particles were well dispersed and hexagonal, with an average size of 12.5–23 nm, corresponding to the FESEM image findings. The selective area electron diffraction (SAED) pattern displays a bright round-concentric ring due to the random orientation of the crystal plane in Figure 6d. The two rings observed on the SAED image represent the diffraction planes (004) and (101) of the fcc structure that allows TiO_2_ in the anatase phase. The results obtained are consistent with the XRD results.

### 3.2. Electrocatalytic Electrochemical Performance

The performance of the PtRu/TiO_2_-GA electrocatalyst as an anode catalyst in DMFC applications was tested using electrochemical analysis. In this section, two key analyses are studied: cyclic voltammetry (CV), which is used to quantify electrocatalytic performance, and chronoamperometry (CA), which was used to clarify the electrochemical long-term stability and durability effects of catalyst poisoning by the surface-adsorbed intermediate species formation during MOR. In a 0.5 M H_2_SO_4_ solution, the CV profiles of all catalysts were measured in the potential range of −0.2 to +1.0 V.

#### 3.2.1. Electrochemically Active Surface Area

Figure 7 depicts the hydrogen adsorption/absorption area in the −0.2 to +1.0 V range. This characterization also included the calculation of electrochemically active surface area (ECSA). In an electrocatalyst, ECSA is a measure of the surface area of PtRu nanoparticles [51]. The technique entailed an electrode current cycle in the voltage range, with charge transfer processes at the activation sites being limited by adsorption. As reactive surface sites for ECSA, the total charge necessary for monolayer adsorption/desorption is employed [52]. As tabulated in Table 3, ECSA for the CV measurement was determined and can be expressed as:(1)$$ECSA(m^{2}{g_{Pt}}^{-1})=\frac{Q}{\Gamma \cdot W_{Pt}}$$
where Q is the charge density or area under the graph ((C) of the CV experiment), Γ (2.1 Cm_Pt_^−2^) is the constant for the charge required to reduce the proton monolayer on the Pt, and W_Pt_ is the Pt loading (g_Pt_) on the electrode. Generally, for the calculation of ECSA for polycrystalline Pt in the H-desorption region, the anodic feed is used rather than the cathodic adsorption charge. This happens because of the difficulty of estimating the adsorption at the overlap of the beginning of hydrogen evolution [53]. The calculation of the PtRu/TiO_2_-GA electrocatalyst appears in ECSA to have the highest value of 68.44 m^2^g_PtRu_^−1^ compared to other electrocatalysts such as PtRu/GA (38.49 m^2^g_PtRu_^−1^), PtRu/TiO_2_ (36.98 m^2^g_PtRu_^−1^), and PtRu/C (20.44 m^2^g_PtRu_^−1^). This is due to several important elements contributing, and the crystallite size of PtRu is one of them, as shown in the PtRu crystallite size for PtRu/TiO_2_-GA, which is the lowest and has the greatest ECSA value, according to Table 1 from the XRD investigation. The catalyst and reaction surface area can both benefit from smaller crystallite sizes. For PtRu/GA, PtRu/TiO_2,_ and PtRu/C, the trend of crystallite size is paralleled by the trend of ECSA value. This demonstrated that there is no agglomeration of PtRu particles in the sample, which has a greater ECSA than PtRu/C. Furthermore, agglomeration might lower the ECSA by reducing the possible surface area to respond. Hence, it can be concluded that a catalyst that achieves a high ECSA value gives a good response to electrocatalyst performance in DMFC applications.

#### 3.2.2. Methanol Oxidation Reaction

As shown in Figure 8, CV was used to assess the electrocatalytic performance of the synthesized electrocatalyst and another electrocatalyst. The electrocatalysts PtRu/TiO_2_-GA, PtRu/GA, PtRu/C, and PtRu/TiO_2_ reached their optimum at room temperature. CV curves were assessed by 0.5 M H_2_SO_4_ in 2M methanol and saturated N_2_ gas. Increasing the H_2_SO_4_ concentration in electrolytes will normally increase ionic conductivity, which will improve the kinetics of electrochemical reactions and, eventually, increase power production and efficiency. In both acidic and alkaline situations, electron transport between the electrode and electrolyte is made possible by the role of H^+^ ions as charge carriers. The pH of the electrolyte can also be changed by adjusting the concentration of H^+^ ions, which can have an effect on the electrode’s reactivity and surface chemistry. An increased H_2_SO_4_ concentration causes the acid to dissolve further, increasing the concentration of H^+^ ions in the electrolyte. Since H+ ions are the main charge carriers in direct methanol fuel cells, this can further speed up electrochemical reaction rates. The potential range of −0.2 to +1.0 V vs. Ag/AgCl is used to measure the various curves. The decreasing order of current density by PtRu/TiO_2_-GA > PtRu/GA > PtRu/TiO_2_ > PtRu/C is illustrated in Figure 8. In comparison to Ag/AgCl, the maximal current density of PtRu/TiO_2_-GA for the MOR appeared to be around 0.7 V. For all of the samples, Table 3 provides the CV data appropriate for peak current density. PtRu/TiO_2_-GA has a current density of 608.17 mA(mg_PtRu_)^−1^, which is 4.29 and 7.69 times greater than PtRu/GA and the commercial electrocatalyst, PtRu/C, respectively.

Generally, reversible electrochemical reactions are characterized by peak currents that are proportional to the square root of scan rates. Peak currents and scan rates may not be correlated linearly or according to the square root relationship for irreversible processes. The peak current’s dependency on both the square root of the diffusion coefficient and the scan rate, as well as their relationship, results in proportionality. The species’ mobility and capacity to reach the electrode surface are determined by the diffusion coefficient, but the amount of time it takes for the species to do so is determined by the scan rate. Since the species has more time to diffuse to the electrode surface at low scan rates, the peak current is higher. Low peak current is caused by the species not having enough time to reach the electrode surface at high scan rates [56]. The TiO_2_-GA catalyst support outperformed the commercial catalyst support in this experiment.

Generally, active reaction sites on the electrocatalyst surface area increase with an increase in overall surface area. The aerogel structure content and the trend match the investigation. A further benefit was the presence of the elevated anatase TiO_2_ composition resulting from the XRD analysis. Generally, anatase has higher electrocatalytic activity than rutile TiO_2_, hence the material combination of PtRu and TiO_2_-GA, where metal–support interaction has a favorable impact on increasing the current density for the electrocatalyst in DMFCs. The aerogel structure of the composite electrocatalyst can improve the total surface area of the electrocatalyst as well as the active reaction site. Additional benefits were the high anatase TiO_2_ contents, as shown by XRD measurements. Anatase has higher electrocatalytic activity than rutile TiO_2_ [57].

PtRu/GA has the second-greatest peak current density with 141.62 mA/mg_PtRu_, which is 1.79 times greater than the commercial electrocatalyst, PtRu/C. The aerogel catalyst support structure is distinguished by PtRu/TiO_2_-GA and PtRu/GA from another electrocatalyst. The capacity of the aerogel boosts catalytic activity by expanding the electrocatalyst surface area. Furthermore, aerogel provides powerful performance in methanol oxidation, as proven by both samples having high peak current densities. The performance of PtRu/C is considerably less than that of PtRu agglomeration, where the crystallite size is determined by XRD and ECSA. The active site surface is a factor that reduces the electrocatalyst’s performance because of this scenario, and the electrocatalyst’s performance is lower as a result. The poor activity of PtRu/TiO_2_ was caused by the poor electrical conductivity of the TiO_2_ catalytic support [4]. For the electrochemical response of catalyst systems, an electroconductive medium was required, as shown by these findings [27].

The histogram in Figure 9 shows the efficiency of using specific activity depending on the catalyst from a specific current normalized to the assessed ECSA. Amazingly, high mass activity can be obtained from PtRu/TiO_2_-GA at approximately 608.17 mA/mg_PtRu_ with the use of specific activity (mA/cm^2^_PtRu_), particularly at almost 0.45 mA/cm^2^ _PtRu_. Consequently, the specific activity of the PtRu catalyst had a high utilization of PtRu mass related to the histogram. This demonstrates that PtRu/TiO_2_-GA has achieved exceptional catalytic activity in electrocatalyst activity. The electrocatalytic activity of the methanol oxidation process was increased as a result of this investigation, which revealed that a higher number of electrons were created, which corresponds with the forward anodic peak current density of PtRu/TiO_2_-GA (30.59 mAcm^−2^). The enhanced electrochemical activity of PtRu/TiO_2_-GA occurs because the PtRu can be linked to distributed and uniform particle sizes as well as higher ECSA values, which result in a large number of electrochemically active sites.

Furthermore, the ratio of forwarding peak current density (I_f_) to backward peak current density (I_b_) was developed to describe the electrocatalyst surface tolerance to the buildup of carbon-containing side products. Voltammetry analyses of the I_f_/I_b_ ratio reveal that PtRu/TiO_2_-GA is 1.09 from cyclic voltammetry. The I_f_/I_b_ value represents the oxidation of methanol in the presence of CO_2_ and the deposition of carbon-derived compounds. Nevertheless, the PtRu catalyst is merely poisoned by the CO intermediate species.

In the CA study, a 2 M methanol solution was treated with 0.5 M H_2_SO_4_ for 3600 s to test the stability and resistance of the electrocatalyst for long-term MOR performance. Figure 10 represents all the PtRu/TiO_2_-GA, PtRu/GA, PtRu/TiO_2_, and PtRu/C electrocatalyst CA curves. All electrocatalysts exhibit rapid degradation, as can be seen in this figure. This may be due to catalytic poisoning caused by the creation of CO_ads_, CH_3_OH_ads_, and CHO_ads_ that normally occur during the methanol oxidation process [58,59]. During the oxidation reaction, as methanol is consumed, the amount of available methanol at the electrode surface reduces, resulting in lower reaction rates and a decrease in oxidation currents. In addition, changes to the surface morphology of the electrode and the formation of passivating layers or other surface species can also impact the oxidation currents by reducing the effective surface area and activity of the electrode. Thereafter, the current density decreases slowly and reaches a (pseudo-steady) state. During the 3600 s, It was observed that the oxidation current densities for PtRu/TiO_2_-GA, PtRu/GA, PtRu/TiO_2_, and PtRu/C were in the order of 5.25 mA/mg > 3.95 mA/mg > 1.45 mA/mg > 1.02 mA/mg, corresponding with the ECSA value and the current density generated in the CV. The PtRu/TiO_2_-GA electrocatalyst shows a higher starting current density and limiting current density compared to the PtRu/C control electrocatalyst.

Based on the CV investigation and comparison of methanol oxidation potential regions throughout the acidic medium, a summary of the overall performance of the produced catalyst can be observed in Table 3. It may be indicated that among other electrocatalysts, PtRu/TiO_2_-GA has the highest peak current density.

#### 3.2.3. DMFC Single Cell Performance

According to the results of the previous half-cell CV, the TiO_2_-GA electrocatalyst performs much better than other electrocatalysts. Further research in a DMFC passive single-cell system was conducted in light of these findings. Figure 11 compares PtRu/TiO_2_-GA and PtRu/C in a single cell. PtRu/TiO_2_-GA showed high power density performance, which was 2.6 times higher than commercial PtRu. The PtRu/TiO_2_-GA electrocatalyst showed a maximum power density of 3.1 mW cm^−2^, while for PtRu/C it was 1.2 mW cm^−2^. The high efficiency of the PtRu/TiO_2_-GA electrocatalyst used in this study was also compared to other commercial PtRu/C electrocatalysts reported by Shimizu and team [60], which had an efficiency of 3.0 mW cm^−2^ and used the same catalyst and passive-mode DMFC. Based on the ease with which CO_2_ molecules form at high current densities in passive-mode DMFC, the passive performance of DMFC may not be as significant as that of an electrocatalyst operating in an active-mode DMFC. The performance of the cell decreases when there is no flow of CO_2_ out of the cell because these molecules can obstruct the catalyst surface sites where the methanol will react. As methanol fuel cannot be controlled in passive mode, the tank is already mixed with methanol by-products, which can block the GDL and lower DMFC performance.

## 4. Conclusions

Finally, a novel 3D hierarchical porous TiO_2_-graphene aerogel (TiO_2_-GA) hybrid catalyst with PtRu/C catalyst wrapped within a 3D hierarchical porous TiO_2_-graphene aerogel framework was developed in this study. The TiO_2_-GA composites were generated using the hydrothermal technique and then freeze-dried before being introduced as a catalytic support for an anodic catalyst in a DMFC. In the electrocatalyst, catalyst supports such as PtRu/TiO_2_-GA, PtRu/GA, PtRu/C, and PtRu/TiO_2_ were synthesized and compared. The PtRu/TiO_2_-GA (608.17 mA(mg_PtRu_)^−1^) electrocatalyst has a high current density, which is 7.69 times greater than the commercial electrocatalyst PtRu/C (79.11 mA(mg_PtRu_)^−1^) according to the findings of this study. The combination of TiO_2_ and aerogel catalyst structures as support catalysts, as well as the metal support face reaction on the PtRu catalysts, gave the catalyst layer new capabilities. Furthermore, in terms of the power density performance based on a DMFC single-cell test, the PtRu/TiO_2_-GA (3.1 mW cm^−2^) revealed superior performance that was 2.6 times higher than that of the PtRu/C (1.2 mW cm^−2^) electrocatalyst. It could be concluded that the high performance of the PtRu/TiO_2_-GA electrocatalyst was due to the potential of a TiO_2_-GA with a large surface area and mesoporous structure, and at the same time, this composite leads to rising catalytic activity performance. Thus, this study effectively demonstrated that PtRu/TiO_2_-GA is the optimum choice for use as anodic catalyst support in DMFCs.

## Figures and Tables

**Figure 1 nanomaterials-13-01819-f001:**
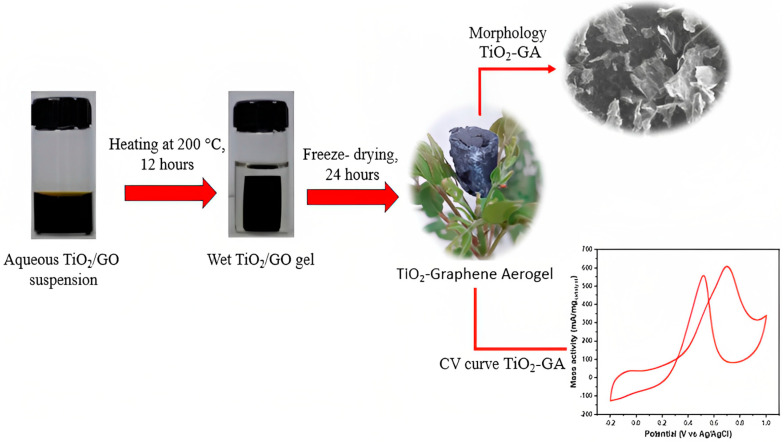
Illustration of the preparation of TiO_2_−graphene aerogel composite.

**Figure 2 nanomaterials-13-01819-f002:**
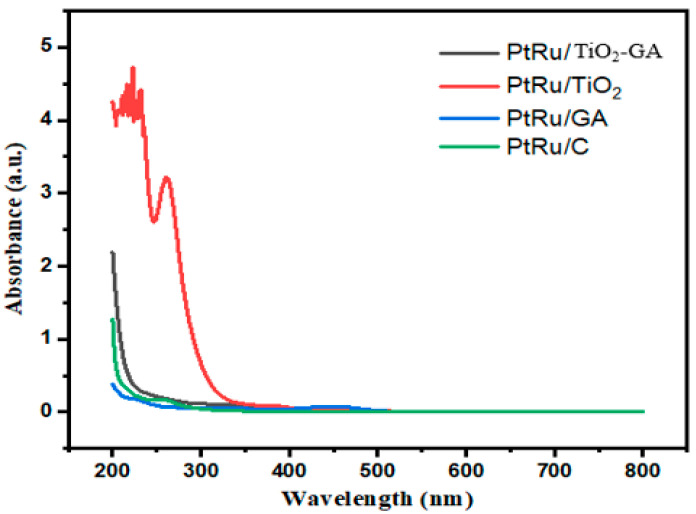
UV/visible spectra of aqueous electrocatalyst.

**Figure 3 nanomaterials-13-01819-f003:**
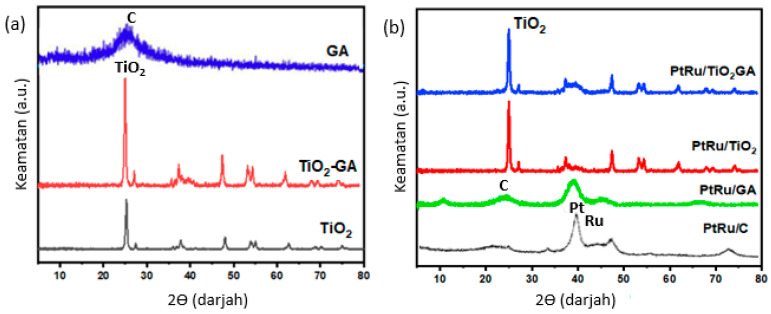
XRD patterns of (**a**) GA, TiO_2_, and TiO_2_-GA composites (**b**) compared electrocatalysts.

**Figure 4 nanomaterials-13-01819-f004:**
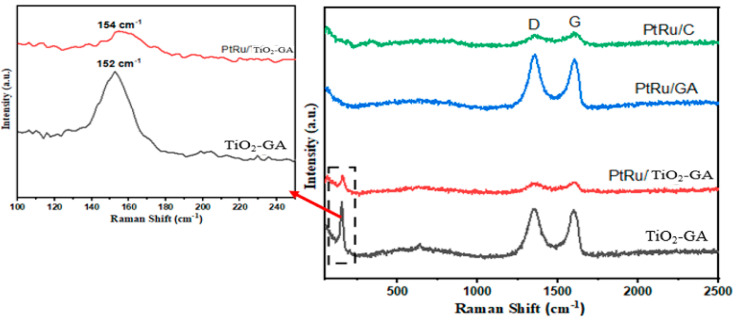
Raman spectra of electrocatalyst.

**Figure 5 nanomaterials-13-01819-f005:**
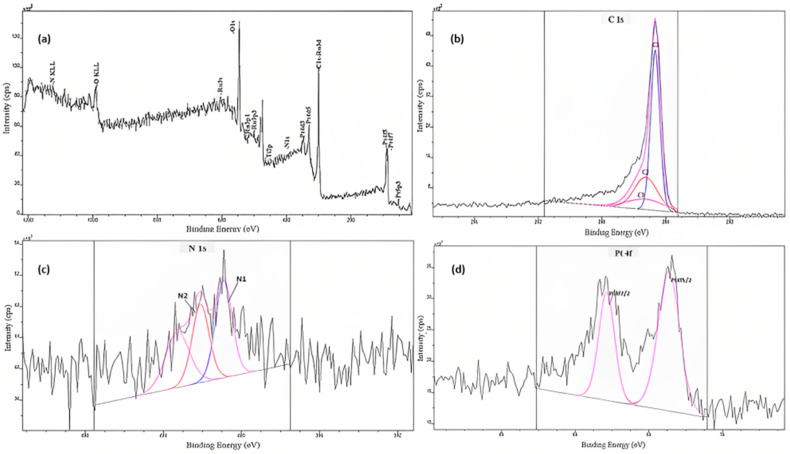
(**a**): The XPS spectrum is broad for the entire PtRu/TiO_2_-GA electrocatalyst; (**b**) the XPS spectrum for element C 1s; (**c**) the XPS spectrum for element N 1s; (**d**) the XPS spectrum for element Pt 4f; (**e**) the XPS spectrum for element Ru 3d; (**f**) the XPS spectrum for element Ti 2p; (**g**) the XPS spectrum for element O 1s.

**Figure 6 nanomaterials-13-01819-f006:**
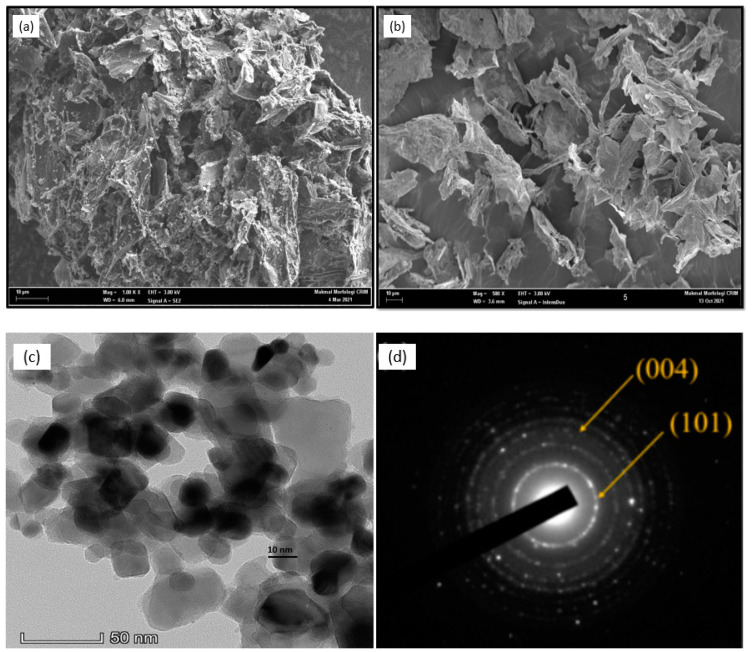
FESEM image of the prepared (**a**) TiO_2_-GA, (**b**) PtRu/TiO_2_-GA, (**c**) TEM image of high performance of PtRu/TiO_2_-GA, and (**d**) SAED pattern marked PtRu/TiO_2_-GA.

**Figure 7 nanomaterials-13-01819-f007:**
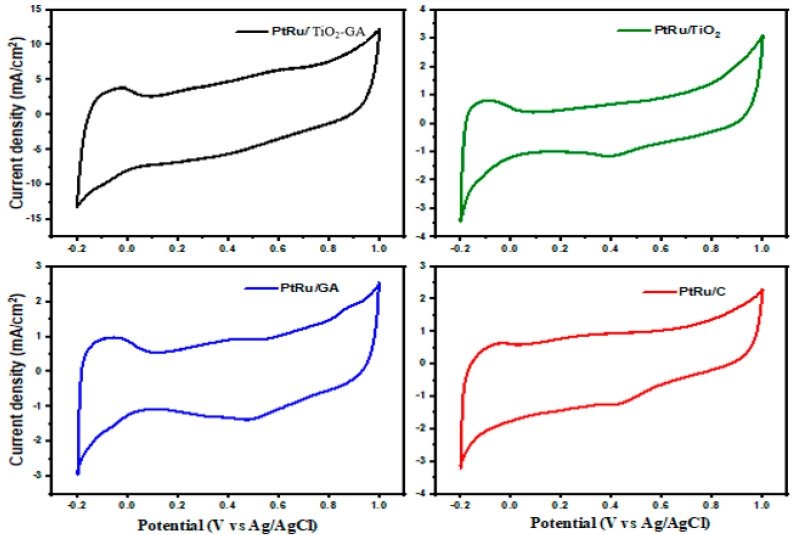
Cyclic voltammetry profiles of the different catalyst support, PtRu/TiO_2_−GA, PtRu/TiO_2_, PtRu/GA, and PtRu/C in 0.5 M H_2_SO_4_ solution at the scan rate of 50 mV s^−1^.

**Figure 8 nanomaterials-13-01819-f008:**
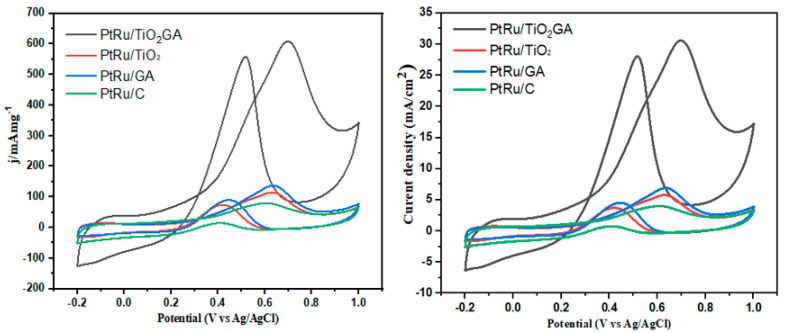
CV curves in 0.5 M H_2_SO_4_ and 2.0 M CH_3_OH aqueous at 50 mV/s.

**Figure 9 nanomaterials-13-01819-f009:**
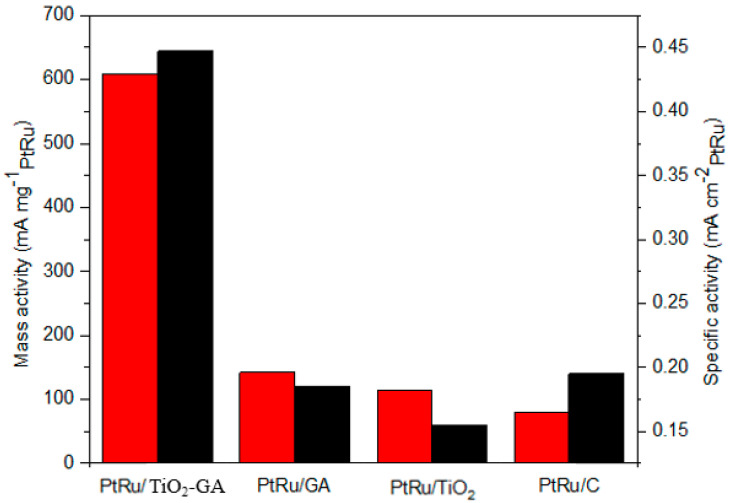
Histogram of mass activity and specific activity electrocatalyst. The graph’s red colour denotes mass activity, which is defined as the current density recorded at a certain voltage where activation losses predominate and normalised by the active material mass loading. Contrarily, the black colour in the graph denotes the specific activity, which is the performance of the samples under consideration as an activity per unit of mass. That number emphasises the efficacy or efficiency of the used substance.

**Figure 10 nanomaterials-13-01819-f010:**
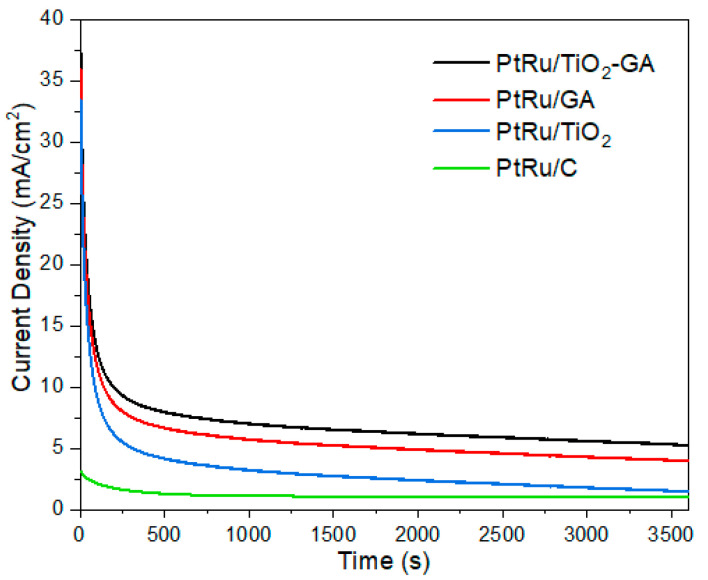
Chronoamperometry curve carried out in 0.5 M H_2_SO_4_ + 2.0 M CH_3_OH.

**Figure 11 nanomaterials-13-01819-f011:**
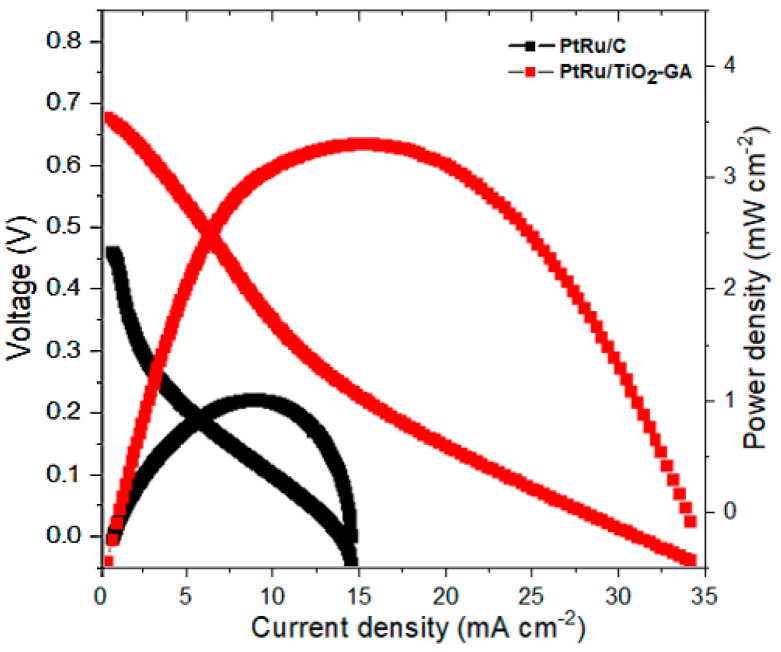
The graph of the I−V (current-voltage) curve from the DMFC single−cell test for the PtRu/TiO_2_−GA and PtRu/C (commercial) electrocatalysts, which were tested in 2.0 M of CH_3_OH at room temperature.

**Table 1 nanomaterials-13-01819-t001:** Estimated crystallite size of electrocatalyst on XRD analysis.

Samples	XRD/Crystallite Size (nm)
PtRu/TiO_2_-GA	3.8
PtRu/GA	7.3
PtRu/TiO_2_	8.1
PtRu/C	9.8

**Table 2 nanomaterials-13-01819-t002:** Results of XPS analysis for PtRu/TiO_2_-GA electrocatalyst.

Element	Atomic Concentration (%)
Pt	11.49
Ru	17.55
Ti	21.3
O	29.95
C	18.84
N	0.87

**Table 3 nanomaterials-13-01819-t003:** Comparison of methanol electro-oxidation on various methods and the performance results with the previous study.

Electrocatalyst	ECSA (m^2^/g_PtRu_)	Peak Potential (V vs. Ag/AgCl)	Onset Potential (V vs. Ag/AgCl)	Mass Activity (mA/mg_PtRu_)	Peak Current Density (mA/cm^2^)	Specific Activity (mA/cm^2^_PtRu_)	Author
PtRu/TiO_2_-GA	68.44	0.7	0.1	608.17	30.59	0.45	This study
PtRu/GA	38.49	0.63	0.19	141.62	6.76	0.18	This study
PtRu/TiO_2_	36.98	0.62	0.17	113.81	5.74	0.15	This study
PtRu/C	20.44	0.61	0.01	79.11	4.02	0.19	This study
PtRu/TiO_2_-CNF	10.4	0.84	0.305	345.64	0.5012	0	Abdullah et al. [27]
	-	0.94	0.6	31	-	-	Basri et al. [26]
PtRu/MWCNT	-	0.91	0.4	326.4	8.66	-	Yen et al. [54]
TiO_2_-PtRu/C	72.4	0.69	0.67	324	-	-	Kolla and Smirnova [20]
PtRu_0_._7_(CeO_2_)0.3/C	4.02	0.19	0.4	21.43	-	-	Guo et al. [55]

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
