# Peer review of "Potential of 3D Hierarchical Porous TiO2-Graphene Aerogel (TiO2-GA) as Electrocatalyst Support for Direct Methanol Fuel Cells"

_nanomaterials, 2023, doi:10.3390/nano13121819_

Round 1
Reviewer 1 Report
Direct Methanol Fuel Cells (DMFCs) are recognized as the most promising sources of energy due to the application of fuel that is characterized by relatively high energy density, low costs, an abundance of production, no complicated storage, and the ability to degrade to CO2. The currently used catalysts are expensive for methanol oxidation, hence there is a demand for alternative catalysts that could address both efficiency and cost factors. Therefore, in this paper, the authors have employed TiO2-based graphene aerogel. Different synthesis/fabrication approaches are employed. The materials are shown to be synthesized by the hydrothermal approach and the catalysts are extensively characterized using various supports. The reported work is suitable for publication after revision.
· Better to avoid acronyms in the abstract. What is ECSA?
· The section introduction is fairly well written. However, Rhodium (Rh) nanocrystals have recently become a hot topic of interest owing to their exceptional methanol oxidation activity as well as greater resistance to byproducts (mainly CO) in acidic/alkaline media. Please address.
· How about depositing on carbonaceous support such as porous carbon?
· Can MnO2 as stated in the literature (doi.org/10.1002/slct.201600867) with larger surface area, active surface sites, and tenacious moisture retention capacity be used as a catalyst? Please discuss.
· The XRD/XPS peaks must be labeled in Figure 3 and Figure 5.
· What is the overall conclusion from the XPS results?
· In Figures 7a-d, and Figure 8 anodic and cathodic peak currents observed in the respective CVs increased with the supported catalyst and the highest for TiO2. At the same time, the anodic peak potentials exhibited a positive shift, whereas the cathodic peak potentials experienced a negative shift. Please explain the fact and determine if the peak currents are proportional to the square root of the scan rates as discussed in the literature; doi.org/10.1016/j.progsolidstchem.2023.100390.
· Is there any effect on the concentration of alkaline H2SO4 electrolytes? What is the role of H cation?
· Please benchmark the current study and its degradation indicators with respect to the recent literature doi.org/10.1021/acsami.0c13755.
· In Figure 10, it can be seen that the methanol oxidation currents on all electrodes gradually decrease as time goes on, what it should be ascribed to? Discussion is missing.
· Tables 3 and 4 can be merged.
· Please summarize the conclusion of section 4 qualitatively by bringing the values to the discussion.
· The plots in Figure 11 look weird.
· A few of the less important XPS plots can either be deleted or moved to supplementary.
A few of the sentences can be improved for clarity.
Reviewer 2 Report
This article reports Potential Of 3D Hierarchical Porous TiO2-Graphene Aerogel (TiO2-GA) As Electrocatalyst Support For Direct Methanol Fuel Cells. Some of the results are interesting but they should be nicely presented. Following revisions are necessary before publications:
1. The abstract should be short and specific highlighting the main approach and finding of the work.
2. The potential use of TiO2 based metal oxide can be explained in introduction sections with following references: Catalysts 2020, 10(5), 546; https://doi.org/10.3390/catal10050546, doi.org/10.1016/j.inoche.2023.110675, doi.org/10.1016/j.est.2023.106713
3. What is the main novelty of this work? Authors used PtRu, how about the cost effectiveness. What is the wt % composition of PtRu in the composites.
4. In XRD, JCPDS file should be updated. The D:G ratio should be mentioned with its impact on the performance of the composites.
5. All the XPS spectrum should be included in same figure. The Fitting of N1s, C1s, Pt 4f, O 1s, Ru 3d and Ti 2p is not correct. It should be refitted with the clear indication of main peaks and satellite peak. Some typographical should not be avoided. For example Ti 2p3/2 and Ti 2p1/2 should be the 3/2 and ½ should be in subscript.
6. The DMFC single-cell test for PtRu/TiO2-GA should be compared with some of TiO2 and PtRu based electocatalysts.
7. Deep literature survey is necessary.
Some grammatical errors and typographical mistakes cannot be avoided. also, some of the sentences were arranged improperly.
Round 2
Reviewer 1 Report
The revised version is suitable for publication.
OK